# Personalised Risk Prediction in Hereditary Breast and Ovarian Cancer: A Protocol for a Multi-Centre Randomised Controlled Trial

**DOI:** 10.3390/cancers14112716

**Published:** 2022-05-31

**Authors:** Stephanie Archer, Nichola Fennell, Ellen Colvin, Rozelle Laquindanum, Meredith Mills, Romy Dennis, Francisca Stutzin Donoso, Rochelle Gold, Alice Fan, Kate Downes, James Ford, Antonis C. Antoniou, Allison W. Kurian, D. Gareth Evans, Marc Tischkowitz

**Affiliations:** 1Primary Care Unit, Department of Public Health and Primary Care, Strangeways Research Laboratory, University of Cambridge, Cambridge CB1 8RN, UK; fsd26@cam.ac.uk; 2Department of Psychology, University of Cambridge, Downing Street, Cambridge CB2 3EB, UK; 3Academic Department of Medical Genetics, University of Cambridge, Addenbrooke’s Hospital, Cambridge CB2 0QQ, UK; nf381@medschl.cam.ac.uk (N.F.); rd643@medschl.cam.ac.uk (R.D.); rochelle.gold@hotmail.co.uk (R.G.); mdt33@medschl.cam.ac.uk (M.T.); 4Manchester Centre for Genomic Medicine, St. Marys Hospital, Oxford Road, Manchester M13 9WL, UK; ellen.colvin@mft.nhs.uk (E.C.); gareth.evans@mft.nhs.uk (D.G.E.); 5Department of Medicine, Stanford University School of Medicine, Stanford, CA 94305, USA; rlaquind@stanford.edu (R.L.); bluett@stanford.edu (M.M.); afan@stanford.edu (A.F.); jmf@stanford.edu (J.F.); akurian@stanford.edu (A.W.K.); 6Cambridge Genomics Laboratory, Cambridge University Hospitals Foundation Trust, Hills Road, Cambridge CB2 0QQ, UK; kate.downes@addenbrookes.nhs.uk; 7Department of Genetics, Stanford University School of Medicine, Stanford, CA 94305, USA; 8Centre for Cancer Genetic Epidemiology, Department of Public Health and Primary Care, Strangeways Research Laboratory, University of Cambridge, Cambridge CB1 8RN, UK; aca20@medschl.cam.ac.uk; 9Department of Epidemiology and Population Health, Stanford University School of Medicine, Stanford, CA 94305, USA; 10Division of Evolution and Genomic Sciences, School of Biological Sciences, Faculty of Biology, Medicine and Health, University of Manchester, Manchester Academic Health Science Centre, Manchester M13 9PL, UK

**Keywords:** personalised risk prediction, breast cancer, epithelial ovarian cancer, CanRisk, polygenic risk scores, genetics

## Abstract

**Simple Summary:**

Women with disease-causing gene changes (faults/mutations) in *BRCA1, BRCA2, PALB2*, *CHEK2* and *ATM* are at an increased risk of developing certain types of cancer—specifically breast (all genes) and epithelial ovarian cancer (only *BRCA1, BRCA2, PALB2*). At present, the risk estimates given to women by most healthcare professionals are broad (e.g., 35–85% lifetime risk of breast cancer for *BRCA1* and *BRCA2*) and are not personalised. This can make it difficult for women to make informed decisions regarding the risk-management options available to them. By combining information about genetic, lifestyle and hormonal risk factors, we can produce a narrower, more personalised risk estimate (e.g., 44% lifetime risk of breast cancer). In this study, we aim to test whether offering personalised risk estimates to women undergoing predictive testing in genetics centres in the UK and USA better supports women’s mental health and choices about their clinical care, relative to standard care. In addition, we will explore the experiences of both staff and women taking part in the study, to understand whether personalised risk estimates are acceptable, feasible and cost-effective for use in clinical care.

**Abstract:**

Women who test positive for an inherited pathogenic/likely pathogenic gene variant in *BRCA1, BRCA2, PALB2*, *CHEK2* and *ATM* are at an increased risk of developing certain types of cancer—specifically breast (all) and epithelial ovarian cancer (only *BRCA1, BRCA2, PALB2*). Women receive broad cancer risk figures that are not personalised (e.g., 44–63% lifetime risk of breast cancer for those with *PALB2*). Broad, non-personalised risk estimates may be problematic for women when they are considering how to manage their risk. Multifactorial-risk-prediction tools have the potential to deliver personalised risk estimates. These may be useful in the patient’s decision-making process and impact uptake of risk-management options. This randomised control trial (registration number to follow), based in genetic centres in the UK and US, will randomise participants on a 1:1 basis to either receive conventional cancer risk estimates, as per routine clinical practice, or to receive a personalised risk estimate. This personalised risk estimate will be calculated using the CanRisk risk prediction tool, which combines the patient’s genetic result, family history and polygenic risk score (PRS), along with hormonal and lifestyle factors. Women’s decision-making around risk management will be monitored using questionnaires, completed at baseline (pre-appointment) and follow-up (one, three and twelve months after receiving their risk assessment). The primary outcome for this study is the type and timing of risk management options (surveillance, chemoprevention, surgery) taken up over the course of the study (i.e., 12 months). The type of risk-management options planned to be taken up in the future (i.e., beyond the end of the study) and the potential impact of personalised risk estimates on women’s psychosocial health will be collected as secondary-outcome measures. This study will also assess the acceptability, feasibility and cost-effectiveness of using personalised risk estimates in clinical care.

## 1. Introduction

Women with pathogenic variants (PVs) in cancer-predisposing genes such as *BRCA1, BRCA2, PALB2*, *CHEK2* and *ATM*, have an increased risk of developing breast cancer (BC), and those with PVs in *BRCA1, BRCA2* and *PALB2* also have an increased risk of epithelial ovarian cancer (EOC), which is the representative histological type of hereditary ovarian cancer for these genes [1]. Current risk estimates given to women for the three breast/ovary genes range from 35–85% lifetime risk for BC and 5–60% lifetime risk for EOC [2]. Furthermore, the risk estimates given to women have a broad range (e.g., 65–79% lifetime risk for BC in *BRCA1*), and do not currently consider personal genetic/lifestyle/hormonal modifiers, even though they are known to influence risk [3,4,5,6,7].

The clinical management of women with high-risk PVs (i.e., *BRCA1/2* and *PALB2*) includes enhanced surveillance, chemoprevention and risk-reducing surgery [8]. For moderate-risk PVs (i.e., *CHEK2* and *ATM)*, the emphasis is more on surveillance and chemoprevention than surgical-risk reduction. Whilst interventions such as chemoprevention, bilateral risk-reducing mastectomy (BRRM) and risk-reducing bilateral salpingo-oophorectomy (RRBSO) can significantly reduce the risks of breast and/or EOC, they require careful consideration, including balancing the risks of developing a potentially life-threatening cancer as well as the potential physical and psychological side effects [9,10,11,12,13,14,15].

Multifactorial-risk-prediction models (e.g., BOADICEA [16,17]) give personalised risk estimates in the general population, in women with cancer family history and those with moderate- and high-risk PVs [18]. These comprehensive risk models combine information about the PV and other genetic and non-genetic modifying factors, such as polygenic risk score (PRS), family history, lifestyle (age at first pregnancy, oral contraceptive pill use, BMI, alcohol consumption) and mammographic density (where available). Clinical tools, such as CanRisk (www.canrisk.org 30 May 2022 [19]), have been designed and tested to support the use of BOADICEA [16] and the associated ovarian-cancer model [17] in clinical practice [20].

Personalised risk estimates may have an important role in supporting women to make informed decisions about their clinical care, as the level of risk (clinical or perceived) has been shown to affect the uptake of risk-management options [21,22,23,24]. In some cases, personalised risk estimates will identify those who might benefit from more intensive early-detection strategies (i.e., those at particularly high lifetime risk, or those with a high cancer risk at an early age). For instance, a *CHEK2* or *ATM* carrier is generally quoted a lifetime risk of 20–30% and not offered BRRM. However, for those carriers with a strong family history of breast cancer and additional risk factors, CanRisk will generally give a lifetime risk of >35%, which would mean discussing BRRM, in accordance with the UK NICE guidelines [25]. Conversely, personalised risk estimates will also identify women with PVs at much lower risk who could then opt to delay or not undergo risk-reducing surgery or chemoprevention. Either way, personalised risk estimates will likely provide healthcare professionals with more nuanced information about their patient’s cancer risk, which may be helpful in the counseling process [14,16].

Whilst multifactorial-risk-prediction tools are now available and recommended for use in clinical practice [25], the availability and impact of personalised risk estimates on women with PVs in moderate/high-risk genes is currently unknown. We anticipate personalised risk estimates may support women to make informed decisions about the uptake of appropriate risk management options, and exploring the potential impact of personalised risk estimates on women’s psychosocial health (i.e., cancer worry, anxiety, quality of life) is also essential. Similarly, evidence on the acceptability, feasibility and economic impact of personalised risk prediction must be collected for them to be widely implemented at pace in clinical practice. With these points in mind, we have designed the randomised controlled trial, described below.

## 2. Materials and Methods

### Study Design

This study is a multicentre randomised control trial across clinical-genetics services based in the UK and the USA. Women who have inherited a familial PV in *BRCA1, BRCA2, PALB2, CHEK2* or *ATM* will be randomised 1:1 into the intervention arm (personalised risk calculation), or the control arm (standard risk calculation), by stratified block randomisation (see section below). See Figure 1, for an overview of the study design.

## 3. Intervention (Personalised Risk Calculation)

Personalised risk calculations will be provided by the CanRisk tool (www.canrisk.org). CanRisk is a multifactorial-cancer-risk-prediction tool that combines genetic (e.g., rare genetic susceptibility variants and common genetic susceptibility variants in terms of a polygenic risk score (PRS)), family history, lifestyle and hormonal risk factors, to calculate the future risk of developing breast and/or epithelial ovarian cancer as well as the risk of being a carrier of a pathogenic variant [16,17]. The predicted cancer risks have been validated in several independent studies [26,27]. The CanRisk tool (see Figure 2, Figure 3 and Figure 4) was originally designed for use by healthcare professionals in a variety of settings [19] and is acceptable for use in clinical practice, particularly in the genetics setting [20].

## 4. Primary and Secondary Outcomes

The primary outcome for this study is both the type and the timing of risk management options (surveillance, chemoprevention, surgery) taken up over the course of the study (i.e., 12 months). The type of risk management options planned to be taken up in the future (i.e., beyond the end of the study) will be collected as a secondary outcome measure. Other secondary outcomes consist of informed decision-making about risk management options (measured by combining objective knowledge, attitude and behaviour [28]), women’s understanding of the test result [29] and the psychosocial impact (including cancer worry [30], anxiety [31] and quality of life [32]). Information on women’s use of health services will also be captured in order to perform a cost-utility analysis. Qualitative research methods offer important insights for the evaluation of health interventions and are increasingly being used as part of RCTs to strengthen the reliability and applicability of experimental findings to real-world settings [33,34,35,36]. As such, we plan to conduct semi-structured qualitative interviews with patients and staff, to better understand the acceptability and implementation of personalised risk calculations in clinical-genetics services.

## 5. Study Setting

This study will take place through clinical-genetics services based in the UK and USA. Initially, this will include Cambridge University Hospital (catchment area of 3,000,000) and Manchester University NHS Foundation Trust (catchment area of 5,000,000) in the UK and Stanford Cancer Institute (catchment area of 8,000,000) in the USA, with the capacity to expand to other clinical-genetic services, if needed. Women will be recruited through each clinical-genetics service and will remain in the clinical-genetics’s pathway throughout their participation in the study.

## 6. Participant Recruitment

Women will be recruited during the triage stage of their referral to clinical genetics; women will have been referred by healthcare professionals from primary care, breast surgery or family history clinics or will have self-referred because a family member has been found to have a PV in *BRCA1, BRCA2, PALB2, CHEK2* or *ATM*. Women will be assessed by the triaging healthcare professional(s), using the eligibility criteria, and flagged to the project coordinator.

## 7. Sample Size

We aim to recruit 1200 participants across the sites over the three years; approximately 600 will be randomised into the study (there is a 50% chance of those recruited having a PV in *BRCA1, BRCA2, PALB2, CHEK2* or *ATM*). We will assess whether there is a significant difference in uptake of RRM, at 12 months. When taking into account the expected distribution of PV carriers in the five genes within our clinical population, the uptake of RRM in *BRCA1/2* carriers and among women in different risk categories [37] and the expected risk distribution among the study participants based on the multifactorial BOADICEA v.6 model (including PRS) [38], the expected uptake of RRM in the intervention arm would be approximately 20%. Therefore:

To detect a difference of 5% in the uptake of RRM between the intervention and control arm, at a power of 80% and a 5% significance level, the sample size required in each arm is 924.

To detect a difference of 7.5% in the uptake of RRM between the intervention and control arm, at a power of 80% and a 5% significance level, the sample size required in each arm is 388.

To detect a difference of 10% in the uptake of RRM between the intervention and control arm, at a power of 80% and a 5% significance level, the sample size required in each arm is 204.

## 8. Eligibility Criteria

Women will be eligible for inclusion if they are: female, aged ≥18, undergoing predictive testing for a PV in *BRCA1, BRCA2, PALB2, ATM* or *CHEK2,* and able to give informed consent. Women will not be eligible for inclusion if they have a previous history of breast cancer or ovarian cancer.

## 9. Randomisation

Women will be randomised 1:1 via block stratification based on their age and the risk classification of the gene (moderate (*ATM, CHEK2*) or high (*BRCA1, BRCA2, PALB2*)). Block-allocation sequences will be computer-generated and shared with a non-participant-facing team member. The next sequence allocation will be shared at the time of each participant’s randomisation, for concealment purposes. It is not possible to blind the allocation group from the research team as additional testing needs to be ordered for women in the intervention arm of the study. Group allocation will not explicitly be shared with the participant, however, it may be inferred from their results letter, where it will either give a range risk figure for breast and ovarian cancer (i.e., 44–63% lifetime risk of breast cancer), or a personalised risk figure (e.g., 45% lifetime risk of breast cancer). The study team does not foresee any circumstances in which a patient would need to be unexpectedly unblinded.

## 10. Study Procedure

Women will be assessed for eligibility at the point of triage when referred to the clinical-genetics service. Once identified as eligible, they will receive an information pack containing details about the study, as well as a link to the online consent form. After completing and submitting the online consent form, the participant will receive a baseline questionnaire (Appendix A), to be completed before attending their standard clinical-genetics appointment. The baseline questionnaire collects data on lifestyle, hormonal factors and family history. It also contains a range of validated scales, measuring numeracy [39], intolerance of uncertainty [40], perceived levels of absolute and comparative risk [41], cancer worry [30], anxiety [31] and quality of life [32]. Participants will also be asked to provide information about their general health and any recent interactions with healthcare providers.

Participants will undergo predictive/pre-symptomatic genetic testing, as per standard clinical procedure, specific for each site. Of those participants whose genetic test shows that they have inherited the familial PV in *BRCA1, BRCA2, PALB2, ATM* or *CHEK2* (approximately 50% of all individuals tested), half will be randomised into the intervention arm of the study, which includes running the polygenic risk panel on their stored DNA sample. Once this result is available, it will be combined with the patient’s clinical-genetics result and their family history of cancer, as well as their lifestyle and hormonal factors using the CanRisk tool. The CanRisk tool will provide a personalised risk estimate which will be shared with the patient via clinical letter. Women who are randomised into the control arm of the study will receive the standard generalised risk estimate at approximately the same time point (post results) as those in the intervention arm of the study.

After getting the final risk assessment via clinical letter, participants will receive follow-up questionnaires (Appendix A), at one, three and twelve months. Questionnaires will collect data on the uptake of risk management options (actual and planned) as well as their knowledge and attitude towards these options. We will also ask participants about their understanding of the test result ([29]; at one month only), perceived levels of absolute and comparative risk [41], cancer worry [30], anxiety [31] and quality of life [32]. As a baseline, participants will also be asked to give information about their general health and any recent interactions with healthcare providers. Participants (*n* = 40; 20 in the UK and 20 in the USA) and staff (*n* = 20; 10 in the USA and 10 in the UK) will be invited to participate in a 30–60 min semi-structured video/audio interview, recorded face-to-face or online (via Zoom/MS Teams). Interviews will explore their experiences of taking part in the study, with a view to understanding whether personalised risk estimates are acceptable and feasible for use in clinical care.

## 11. Data Analysis

Descriptive statistics will be used to summarise participants’s demographic characteristics. Univariate and multivariable analyses (linear and logistic regression) will be applied, taking into account the study arm and place of care (UK vs. USA). The multivariable models will adjust for the demographic characteristics reported at the beginning of the study. Multilevel models will be considered to investigate differences attributed to specific participant characteristics.

Health economic analysis will take the form of a lifetime cost-utility analysis, with health outcomes expressed as quality-adjusted life-years (QALYs), conforming to methods recommended by NICE [42]. Changes in health-related quality-of-life will be measured using the SF-6D derived from the SF-12 data [32], and modeling will be used to extrapolate lifetime costs and benefits.

Qualitative data collected through semi-structured interviews with patients and staff will be analysed deductively using thematic analysis [43], drawing from Sekhon’s Theoretical Framework of Acceptability (TFA; [44]) and the Consolidated Framework for Implementation Research (CIFR; [45]). Patient and staff data will be analysed separately but data from the UK and USA will be combined in order to compare and contrast between the different health settings.

## 12. Discussion

As we progress further into the genomic era, the increased demand for—and accessibility of—genetic testing highlights a growing need to improve the pace, quality and accuracy of information shared with patients. With the development and availability of PRS and multifactorial-risk-prediction tools, we expect the current “one size fits all” approach to become increasingly outmoded in clinical-genetics practice. By combining a patient’s genetic test result, along with their family history, PRS and lifestyle/hormonal factors (and, where available, mammographic density), a more precise cancer-risk estimate can be generated in a relatively short and clinically useful timescale.

By studying the integration of personalised risk estimates into the existing clinical-genetic-testing process, as well as for a year beyond the testing process, we will be in a better position to understand how personalised risks support and inform women’s decision-making processes as well as their wider impact on women’s psychosocial health. Moreover, we will be able to collate information on the acceptability, feasibility and economic impact of personalised risk prediction, which will be required for future studies and/or widespread adoption within clinical settings.

Whilst multifactorial-risk-prediction tools are becoming more widely recommended for stratifying risk [25], testing for PRS in clinical-genetics services is in an introductory phase [46] and its clinical utility through the disease trajectory is yet to be established [47]. As our study is one of the first translational research projects to determine risk in healthy women [48,49], our findings will help researchers and clinicians to better understand the potential for the use of and PRS in clinical settings, as well as the challenges in doing so. Furthermore, knowledge gained from this study will be essential to develop new and urgently needed frameworks for the responsible use of PRS in clinical practice [48].

## Figures and Tables

**Figure 1 cancers-14-02716-f001:**
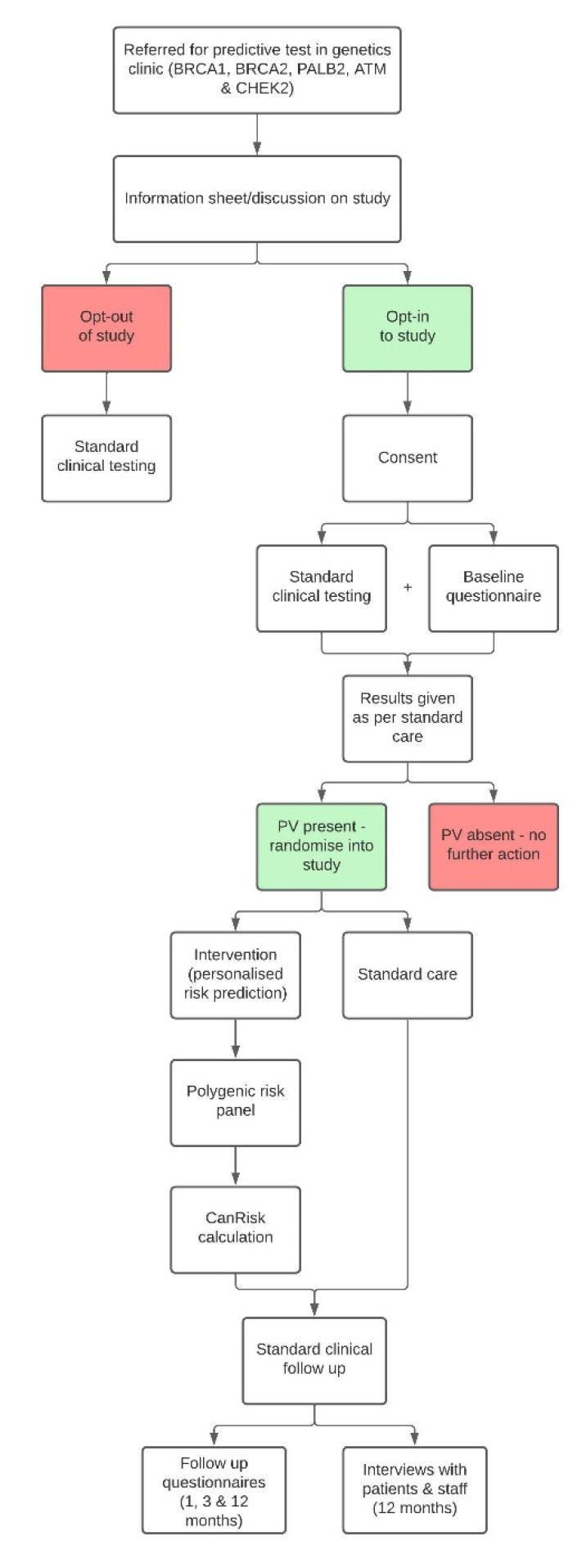
Overview of the study design.

**Figure 2 cancers-14-02716-f002:**
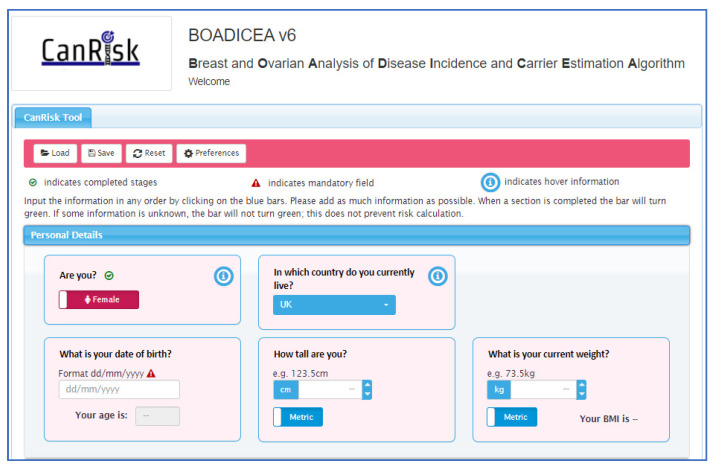
CanRisk tool, detailed view of one topic area.

**Figure 3 cancers-14-02716-f003:**
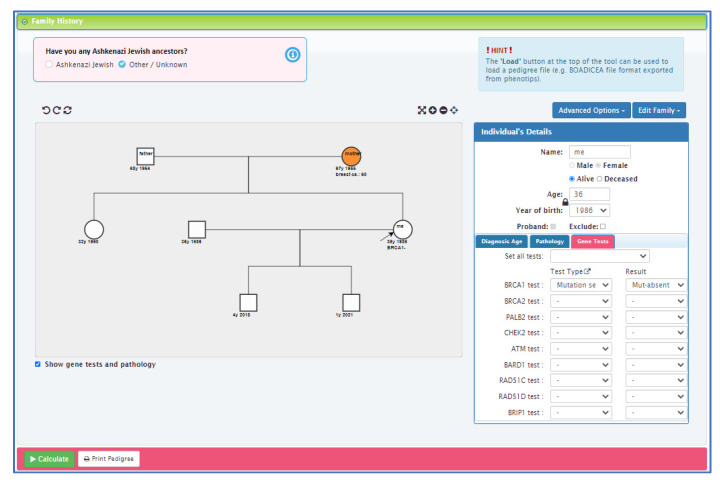
CanRisk tool, detailed view of the interactive graphical pedigree editor.

**Figure 4 cancers-14-02716-f004:**
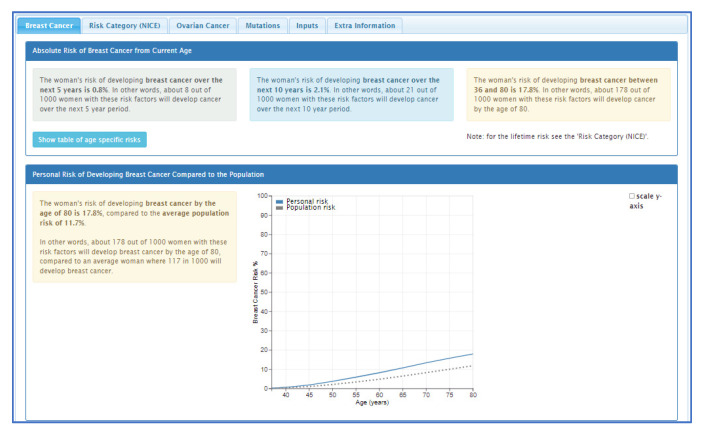
CanRisk tool, results section (post risk calculation).

## Data Availability

Not applicable.

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
