# Peer review of "Personalised Risk Prediction in Hereditary Breast and Ovarian Cancer: A Protocol for a Multi-Centre Randomised Controlled Trial"

_cancers, 2022, doi:10.3390/cancers14112716_

Round 1

Reviewer 1 Report

Interesting research protocol. But clearly it is not an original research manuscript in itself since there are no results. 

let's concentrate on relevant scientific considerations: 
The manuscript is interesting, well, written and scientifically sound. The topic covered is highly relevant in clinical pratice. The research project described is therefore of high interest, and I can only look forward to the results in a few years. And the teams involved are all prestigious cancer genetics teams. 

I only have minor comments: 

INTRO
Paragraph 1, line 2: ATM could actually also be an ovarian cancer susceptibility gene, a list of references is available here: https://ascopubs.org/doi/10.1200/OP.21.00382

Paragraph 4
"In some cases, personalised risk estimates will identify those who will benefit from more intensive early detection strategies". I would be more cautious and say might benefit, as the use of CanRisk to influence PV carrier management is not entirely clinically validated as far as I know. 

METHODS
Participant recruitment and sample size: maybe you should present here the two recruitment stages, opt-in and then randomizazion if PV. And specify that the 600 participants would be those who made it to the second stage (i.e. PV carriers). If I get it right. 

Reviewer 2 Report

The aim of this manuscript is to evaluate whether providing personalized risk estimates, to women subjected to predictive testing in genetics centre in UK and USA, improves and supports women’s mental health and choices, about their clinical care. Moreover, the authors also analyze the experiences of both staff and women, taking part in the study, to understand if personalized risk estimates are acceptable for use in clinical care.

This manuscript shows rich content, providing a deep insight for some works: I found it to be well-written and accessible, providing sufficient information on this topic. This is the additional point, which makes this manuscript original, in comparison to published literature. Even if the manuscript provides an organic overview, with a densely organized structure and based on well-synthetized evidence, there are aspects to be mentioned, to make the article fully readable. For these reasons, the manuscript requires minor changes.

Please find below an enumerated list of comments on my review of the manuscript:

INTRODUCTION:

LINE 66: Epithelial ovarian cancer (EOC) is considered the most diffused type of ovarian cancer. As suggested by several and recent studies (see, for reference: Giusti, I., Bianchi, S., Nottola, S. A., Macchiarelli, G., Dolo, V. (2019). CLINICAL ELECTRON MICROSCOPY IN THE STUDY OF HUMAN OVARIAN TISSUES. EuroMediterranean Biomedical Journal, 14), in the heterogenous group of this neoplasia, epithelial ovarian cancer is the most representative (90%), characterized by several cellular subtypes (mucinous, serous, endometrioid, and clear cell). In this context, the manuscript may benefit from providing this information, achieving a balance of detail for non-expert readers and those with more expertise in the field.

LINE 83: Due to the fact that a single variant is not informative for determining the disease risk of a polygenic disorder, Polygenic Risk Score is informative, as it provides prognostic information on disease course, as suggested by several and recent studies (see, for reference: Lewis, C. M., & Vassos, E. (2020). Polygenic risk scores: from research tools to clinical instruments. Genome medicine12(1), 1-11). For these reasons, the authors should highlight this aspect, in order to complete the information and provide to the readers recent evidence on this topic.

As regards the main topic, it is interesting and certainly of great scientific and clinical impact: in fact, this manuscript touches a significant area, by analyzing the effects of personalized risk prediction in hereditary breast and ovarian cancer. As regards the originality and strenghts of this manuscript, this is a significant contribute to the ongoing research on this topic. Overall, the contents are rich, and the authors also give their deep insight for some works.

As regards the section of methods, there is a specific and detalied explanation for the majority of methods used in this study: this is particularly significant, since the manuscript relies on a multitude of methodological and statistical analysis, to derive its conclusions. The methodology applied is overall correct, the results are reliable and adequately discussed.

The conclusion of this manuscript is perfectly in line with the main purpouse of the paper: the authors have designed and conducted the study properly. As regards the conclusions, they are well written and show an adequate balance between the description of previous findings and the results presented by the authors.

Finally, this manuscript also presents a basic structure, properly divided and characterized by organic and detailed figures and tables. This manuscript looks like very informative since there are few evidence on this topic. As regards tables and figures, they are legible and easy to follow.

In conclusion, this manuscript is densely presented and well organized, based on well-synthetized evidences. The authors were lucid in their style of writing, making it easy to read and understand the message, portrayed in the manuscript. Besides, the methodology design was rigorous and appropriately implemented within the study. However, many of the topics are very concisely covered. This manuscript provided a comprehensive analysis of current knowledge in this field. Moreover, this research have futuristic importance and could be potential for future research. However, the minor concern of this manuscript is with the introductive section: for these reasons, I have minor comments only for the introductive section, for improvement before acceptance for publication. The article is accurate and provides relevant information on the topic and I suggest minor changes to be made in order to maximize its scientific impact. I would accept this manuscript, if the comments are addressed properly.
